# Primary, Bilateral and Diffuse Renal Non-Hodgkin’s Lymphoma in a Young Woman Suffering from Turner Syndrome

**DOI:** 10.3390/jpm11070644

**Published:** 2021-07-07

**Authors:** Bernardo Rossini, Tetiana Skrypets, Carla Minoia, Angela Maria Quinto, Gian Maria Zaccaria, Cristina Ferrari, Nicola Maggialetti, Alessandro Mastrorosa, Pietro Gatti, Michela Casiello, Sabino Ciavarella, Attilio Guarini

**Affiliations:** 1Hematology Unit, IRCCS Istituto Tumori “Giovanni Paolo II”, Viale O. Flacco, 70124 Bari, Italy; b.rossini@oncologico.bari.it (B.R.); tetianaskrypets@gmail.com (T.S.); quintoangelam@gmail.it (A.M.Q.); gianmaria.zaccaria@gmail.com (G.M.Z.); sabinociavarella@yahoo.it (S.C.); attilioguarini@oncologico.bari.it (A.G.); 2Clinical and Experimental Medicine PhD Program, University of Modena and Reggio Emilia, Via Università, 4, 41121 Modena, Italy; 3Nuclear Medicine Unit, Interdisciplinary Department of Medicine, University of Bari “Aldo Moro”, Piazza Giulio Cesare 11, 70124 Bari, Italy; ferrari_cristina@inwind.it; 4Department of Medical Science, Neuroscience and Sensory Organs, University of Bari “Aldo Moro”, Piazza Giulio Cesare 11, 70124 Bari, Italy; n.maggialetti@gmail.com; 5Urology Unit, IRCCS Istituto Tumori “Giovanni Paolo II”, Viale O. Flacco, 70124 Bari, Italy; a.mastrorosa@oncologico.bari.it; 6Internal Medicine Unit, “A. Perrino” Hospital, Strada Statale 7 per Mesagne, 72100 Brindisi, Italy; pietro.gatti@asl.brindisi.it; 7Pathology Unit, “A. Perrino” Hospital, Strada Statale 7 per Mesagne, 72100 Brindisi, Italy; michelacasiello@gmail.com

**Keywords:** primary renal lymphoma, non-Hodgkin’s lymphoma, diffuse large B cell lymphoma, Turner syndrome, personalized therapy

## Abstract

Primary renal lymphoma (PRL) is a rare form of non-Hodgkin’s lymphoma (NHL) restricted to and primarily involving one or both kidneys, with no lymph node extension. It accounts for <1% of extranodal lymphomas, and descriptions in the literature are limited. Here, we describe an unprecedented case of bilateral PRL in a 44-year-old woman with Turner syndrome and discuss both diagnostic and therapeutic issues in the light of the available literature in the field. A personalized approach to this rare disease is necessary.

## 1. Introduction

Primary renal lymphoma (PRL) is defined as an extranodal non-Hodgkin’s lymphoma (NHL) primarily involving one or both kidneys, with no additional nodal or extranodal localization. The etiology of the disease remains unknown since the kidney is neither a lymphatic organ nor shows physiologically any trace of organized lymphatic tissue. Owing to its rarity, information on diagnosis, treatment and prognosis of this condition remains limited. PRL accounts for approximately 0.7–1% of all non-Hodgkin’s lymphomas (NHL), and diffuse large B cell lymphoma (DLBCL) represents the most prevalent pathological histotype [1]. The median age at diagnosis ranges from 60 to 70 years, with the prevalence in men [2,3]. Conversely, secondary renal involvement is more frequent and could occur in 30–60% of all NHL [4].

PRL usually requires a challenging differential diagnosis with renal cell carcinoma. Typical symptoms of PRL include flank pain, hematuria, weight loss, fever and potential signs of acute or chronic renal failure [4,5,6]. PRL has a more frequent monolateral involvement, although bilateral kidney localization could be observed in up to 7.9–20% of cases [3,7]. The disease is typically associated with rapid progression and dismal prognosis, showing a median survival rate of nearly 1 year [3]. In a population-based study from the Surveillance, Epidemiology, and End Results (SEER) database, the 1-year and 5-year relative survival (RS) rates were 78% and 64%, respectively. An amelioration of OS was observed in the period 2000–2013 in comparison to 1980–1999 due to the introduction of immunotherapy in association with chemotherapy [3].

The first case of PRL was reported by Coggins et al. nearly 40 years ago [8] and since then, few other cases have been described, the majority of them with monolateral localization [2,3,9]. Among these cases, to our knowledge, none have been reported in association with rare chromosomal syndromes. Here, we present the case of bilateral PRL in a patient affected by Turner syndrome (TS). It is well-known that TS patients have a higher risk of developing solid cancers, especially central nervous system tumors and melanoma [10]. However, few data are available on the incidence and association of hematological malignancies in this subset of patients [10,11,12]. The authors present this clinical case aiming to describe (i) a rare association between PRL and a chromosomal syndrome predisposing to cancer, (ii) a personalized diagnostic and therapeutic approach and (iii) the response to first-line treatment.

## 2. Case Report

In December 2020, a 44-year-old woman diagnosed with Turner Syndrome (karyotype 45, X0) started undergoing examinations for flank pain which had been developing over the previous two months. Ultrasound scanning and computed tomography (CT) of the abdomen showed multiple hypodense nodular lesions with definite boundaries within the cortical area of both kidneys (Figure 1a,b). Neither mesenteric nor retroperitoneal lymphadenopathies were noticeable, and the spleen volume was normal. The renal pelvis appeared clear with no signs of infiltration. Such evidence was suggestive of a lymphoproliferative process, though no systemic B symptoms were referred. Patient comorbidities included hypertension, hypothyroidism post-thyroiditis and obesity (body mass index (BMI), 30.3). The oncologic family history was negative.

She completed the diagnostic workup with blood cell count, blood chemistry and hepatitis and HIV serology, which were within the normal limits, except for a known hypertransaminasemia (two times more transaminases than is normal), but a conclusive diagnosis to explain this alteration was not possible, and differential diagnosis between nonalcoholic fatty liver disease (NAFLD) (cholesterol, 273 mg/dL; high-density lipoproteins (HDL), 54 mg/dl; obesity) and autoimmune liver disorder, though autoimmunity was negative (antinuclear antibodies (ANA), anti-smooth muscle antibodies (ASMA), anti-liver/kidney microsomal antibodies (anti-LKM), antimitochondrial antibodies (AMA)), was taken into consideration. It is to be considered that Turner syndrome often manifests with hypertransaminasemia whose cause is not always documentable and could be related to morphostructural congenital hepatic alterations [13,14]. Hepatic alteration in TS most commonly manifests as asymptomatic hypertransaminasemia and, while in many cases liver injury does not progress to cirrhosis, there is a six fold increased risk of cirrhosis in patients with TS compared with the general population [15,16]. Bone marrow biopsy and immunophenotyping excluded lymphomatous infiltration. Positron emission tomography/CT (PET/CT) revealed multiple areas of intense and focal [^18^F]FDG uptake in both kidneys (maximum standardized uptake value (SUV), 20.2) consistent with confluent nodular lesions detected by CT, with no washout modification in the delayed image acquisition (Figure 2). No additional areas of pathological [^18^F]FDG uptake were found.

Based on this evidence, the patient was a candidate for contrast enhancement ultrasound (CEUS)-guided renal biopsy. Histological examination of the renal tissue specimen showed remarkable lymphocyte infiltration consisting of medium-to-large lymphoid cells replacing the normal renal parenchyma. Through immunohistochemistry, malignant cells appeared LCA^+^, CKMNF-116^−^ (positive in the ductal epithelial component), CD20^+^, CD3^+^, BCL6^+^ (>30%), CD10^+^, CD5^−^, cyclin D1^−^, BCL2^−^ and MUM1^+^, with a proliferation rate (Ki-67) of approximately 70% (Figure 3). Moreover, fluorescence in situ hybridization (FISH) analysis for C-MYC and BCL2 showed negative expression. The pathological report was conclusive for a diagnosis of DLBCL, not otherwise specified (DLBCL, NOS), according to the latest revised WHO classification [17]. According to morphology, immunohistochemistry and FISH, differential diagnosis with high-grade B cell lymphoma, double-hit lymphoma and lymphoblastic lymphoma was excluded. The final stage according to the Lugano criteria was IV [18] and the International Prognostic Index (IPI) reflected a high-risk disease. The central nervous system (CNS)-IPI [19] results were also high.

The patient then started immunochemotherapy according to the R-CHOP regimen (rituximab, cyclophosphamide, adriamycin, vincristine and prednisone). The first cycle was administered over 3 days instead of one in order to avoid potential renal toxicity. Six courses were planned and completed without significant bone marrow, hepatic and renal toxicity. Prophylaxis for secondary CNS lesions was also performed with intrathecal methotrexate and aracytin while high-dose methotrexate had been planned at the end of induction therapy. The final PET/CT assessment showed persistence of the disease in nodular lesions of the parenchyma of both kidneys, with a reduction of their number and metabolic activity (maximum SUV, 5.7) compared to the initial report. The Deauville score (DS) was 4 [20]. Taking into account the final PET/CT result, the aggressive disease behavior and the rare nature of the localization, after multidisciplinary consultation, salvage approach was undertaken. The R-DHAOx regimen (rituximab dexamethasone, high-dose aracytin, oxaliplatin) which was chosen as the second-line treatment and consolidation with an autologous hematopoietic stem cell transplant (ASCT) were planned.

## 3. Discussion

Here, we report the first case of PRL in a patient with TS. TS is an X-linked disorder affecting about 1 in 2000 live female births, characterized by a complete or partial monosomy of the X chromosome. Patients with TS have increased gonadotropin concentration from infancy or older age, usually the age typical for the start of puberty in healthy girls, and low titers of estrogens. Hypergonadotropic hypogonadism is a typical TS feature in adolescence and adulthood. A number of studies has reported recurrent association of TS with congenital diseases, renal and genitourinary anomalies and autoimmune disorders. In a retrospective study performed at the Mayo Clinic from 1950 to 2017, 317 patients with TS were shown to feature a threefold increased mortality over the aged-matched controls, and the leading causes of death were cardiovascular and liver diseases, as well as malignancies [21].

The risk of cancer in TS has been only partially studied, although chromosomal and hormonal abnormalities themselves might impact significantly the probability of cancer development. However, while an increased risk of solid tumors in TS has been recognized, the occurrence of hematologic malignancies is rarely reported [11,12,13]. Over a cohort of 3425 women in Great Britain observed between 1959 and 2002, an increased risk of gonadoblastoma, meningioma, brain tumors, bladder and corpus uteri cancer as well as melanoma was described, but only one NHL case and one Hodgkin lymphoma case were mentioned [11]. Similarly, a study using the Swedish Cancer Registry identified 1409 women with TS whose overall risk of solid cancer was 1.34-fold higher than in the general population [10,21] while the risk of hematological malignancies appeared increased only in subjects with Klinefelter syndrome.

With regardto the epidemiology and clinical features of PRL, helpful information can be found in two recent large case series including 723 and 599 patients, respectively [2,3]. The first was a population-based study from the SEER database. Among the 723 PRL patients, the most common histotype of lymphoproliferative neoplasm was DLBCL, with the incidence of 0.053/100,000 person-years and the mortality of 0.036/100,000 person-years. The incidence of PRL was increasing significantly with an annual percentage change of 3.45% (*p* < 0.001). The demographic and clinical characteristics of those patients consisted in the median age of 63.7 years, mostly male, with prevalent unilateral involvement and bilateral presentation in only 7.9% of cases. Younger patients (under 18 years of age at diagnosis) had bilateral involvement more frequently [3]. The second series described demographic, clinical and pathological characteristics of PRL, as well as factors affecting the survival of 599 patients in the SEER database from 1973 to 2015 [2]. In that case series, the age-adjusted incidence rate of PRL was 0.035/100,000 and an increasing trend was observed with an annual percentage of 3.3%.

The etiology of PRL is not completely understood. Patients with immunosuppression have a higher incidence of lymphoproliferative disorders, although only isolated cases of PRL are reported in HIV-positive patients. The loss of chromosome X in TS could favor the onset of PRL as it contains nearly 10% of all microRNAs with the function implicated in controlling immunity and cancer [10]. TS-affected women or subjects with monosomy of the chromosome X are reported to have a higher risk of autoimmune diseases, which in turn may increase the risk of cancer. While it is common for NHL to affect the kidneys, PRL is an interesting and unique condition involving the renal parenchyma which lacks lymphoid tissue. A number of theories, in fact, have been traced to explain its pathogenesis [4], such as (i) the “inflammatory nidus theory” (also implicated in lymphomas associated with chronic thyroiditis and *Helicobacter pylori* gastritis), suggesting that chronic inflammation fosters oncogenic events in tissue-infiltrating lymphoid cells; reports of PRL in patients with chronic pyelonephritis support this theory, although PRL is known to occur in patients with no primary kidney disease; (ii) the association of PRL with other chronic inflammatory and infectious diseases such as Sjogren’s disease, systemic lupus erythematosus and Epstein-Barr virus infections; (iii) PRL originates in the lymphatics surrounding the renal capsule and further invades the kidney as solitary or multiple focal masses with unilateral or bilateral extension [2]. In our rare and bilateral case of PRL, beyond TS, the patient had a tendency to inflammation-driven conditions including hypothyroidism post-thyroiditis. The “inflammatory nidus theory” could be thus hypothesized as the etiopathogenesis of the lymphoma in our patient, but we do not have sufficient information to prove this assumption.

Concerning the treatment approach to PRL, for monolateral localization, some cases treated using surgery or surgery plus chemotherapy have been reported [2,9]. Due to the rarity of the disease, no evidence emerged from clinical trials; thus, the gold standard therapy for the corresponding histotype is indicated. In our patient, a treatment according to the guidelines for first- and second-line DLBCL therapy was performed [22,23].

In conclusion, despite the continual increase of incidence and single case records of PRL worldwide, the disease remains rare and offers many challenges for diagnosis and treatment. This is the first report describing its association with a genetic condition such as TS. The pathogenesis of the disease is still unknown, but in the association with TS, a role of altered immune response and chronic inflammation could be discussed. The improvement of clinical practice with the common use of targeted drugs (e.g., rituximab) and CNS prophylaxis provides progressive amelioration of outcomes for these patients. Additional studies are warranted to define the PRL etiology and design tailored algorithms of diagnosis and therapy for these patients. The case could be of practical help to manage cases of bilateral PRL and highlight the possible increased risk of malignancies in TS-affected persons.

## Figures and Tables

**Figure 1 jpm-11-00644-f001:**
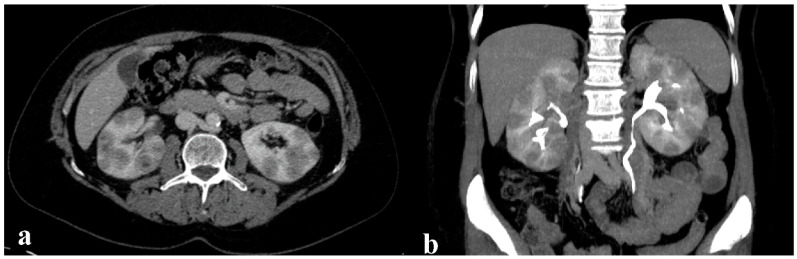
CT imaging placing the suspicion of the lymphoproliferative disorder. (**a**) Axial acquisition after the contrast medium injection: multiple hypodense nodular masses with defined margins are clearly evident in both kidneys in the cortical area. Renal veins are patent with an occasional accessory retroaortic left renal vein. Neither mesenteric nor retroperitoneal adenomegalies are evident. (**b**) MPR reformatting of excretory phase: typically, the renal pelvis is well-defined without infiltrations signs. Note: spleen volumetry is normal.

**Figure 2 jpm-11-00644-f002:**
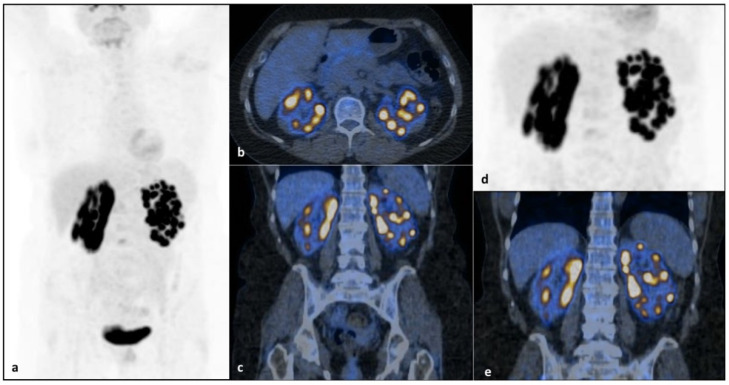
PET/CT imaging revealed multiple areas of intense and focal [^18^F]FDG uptake in both kidneys at diagnosis (maximum SUV, 20.2). [^18^F]FDG PET/CTmaximum intensity projection (MIP) (**a**), axial (**b**) and coronal (**c**) fused PET/CT images reveal multiple areas of focal intense [^18^F]FDG uptake in both kidneys, corresponding to confluent CT-detected nodular lesions without any washout modification on the delayed focused images (**d**,**e**). Of note, despite the limit represented by radioactive urine interference in the kidneys in this patient, performing a delayed acquisition with or without diuretic administration could help in discriminating physiological uptake from the pathological one.No further areas of pathological [^18^F]FDG uptake were found in the remaining body segments. Radiopharmaceutical accumulation in the bladder is referred to physiological [^18^F]FDG excretion (**a**).

**Figure 3 jpm-11-00644-f003:**
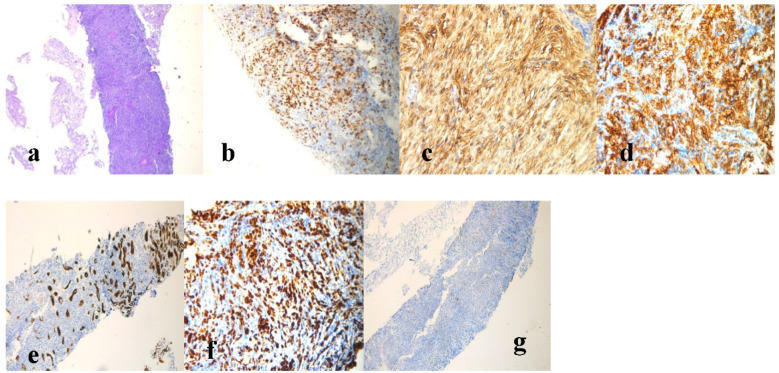
Histological examination of the renal specimen and immunohistochemistry staining were consistent with the diagnosis of DLBCL.(**a**) Hematoxylin and eosin (HE)staining; (**b**) BCL6 staining. (**c**) CD10 staining (20× magnification); (**d**) CD20 staining (20× magnification); (**e**) CKMNF116 staining; (**f**) Ki-67 staining (20× magnification); (**g**) CD3 staining (4× magnification).

## Data Availability

Not applicable.

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
