# Peer review of "Primary, Bilateral and Diffuse Renal Non-Hodgkin’s Lymphoma in a Young Woman Suffering from Turner Syndrome"

_jpm, 2021, doi:10.3390/jpm11070644_

Round 1

Reviewer 1 Report

The changes have improved the manuscript and I do not have additional comments.

Author Response

The Authors thank for the positive comment. 

Reviewer 2 Report

This revised version of the manuscript by Rossini B. et al. fully addresses the reviewer's comments.

No further requests for editing are raised.

Careful checking of the English language and editing is suggested.

Author Response

The authors thank for the positive comments. 

An edition of English language has been performed. 

Reviewer 3 Report

Please see the attached review.

Author Response

16-6-21

Dear Reviewer 3,

the Authors thank you for the supplementary comments, to which they tried to respond at their best.

The manuscript has been reviewed by a native English speaker.

Response to question 1

The Authors have modified the sentence in the Introduction by adding “Among these cases, to our knowledge, ..”.

Response to Question 2

We have now added the advised reference

13 - Manola, K.N.; Sambani, C.; Karakasis, D.; Kalliakosta, G.; Harhalakis, N.; Papaioannou, M. Leukemias associated with Turner syndrome: report of three cases and review of the literature. Leuk Res. 2008;32(3):481-6. doi: 10.1016/j.leukres.2007.06.004

and cited the paper 16-17 in the Introduction, now numbered as 11-12.

Response to Question 3

We have checked for the requested evaluations and here we present the reports:

- thyroid us (01/07/2020) and endocrinologic evaluation:  hypothyroidism post-thyroiditis. ECT: thyroid gland reduced in volume (right 8*12*30 mm, left 7*14*27), diffusely inhomogeneous and hypo echoic echo structure for chronic thyroiditis. Absence of nodular formations.

Patient on therapy with levotiroxine 125 ug/die from Monday to Friday, 150 ug/die on Saturday and Sunday.

- thyroid autoimmunity: TSH 8.49 uUI, FT4 1.88 ng, FT3 1.82 ng. Recent autoimmunity exams are not available.

The text in the case presentation (page 2, paragraph 2) has been changed as follows:

“Patient comorbidities included hypertension and hypothyroidism post-thyroiditis.”

- liver us (28/09/2018): liver with a volume within the normal limits, with an echostructure with mild steatosis, free from nodular lesions.

- liver autoimmunity & exams: HCV neg, HBV neg, bil 0.73, AST 63, ALT 106, GGT 133, ANA neg, AMA neg, ASMA neg, anti-dsDNA neg, anti-LKM neg.

Patient on therapy with urso desossicolic acid 300 mg x 3 /die

According to these exams, we have changed the text as follows (page 2, paragraph 3):

“A known hypertransaminasemia was confirmed (transaminases 2 folds higher), but a conclusive diagnosis of autoimmune liver disorder associated with the genetic condition was not possible due to the negativity of autoimmunity tests (antinuclear antibodies - ANA, anti-smooth muscle antibodies - ASMA, anti–liver-kidney microsomal antibody – anti-LKM, anti-mithocondria antibodies – AMA).”

Response to Question 4

The Authors are sorry but they cannot address this question. We prefer not to restore the sentence about pregnancy because it contains very personal elements in a patient who is however easily identifiable due to the rarity of the disease and the affiliation of the center.

Response to Question 5

Thank you for modifications. We have rewritten the text as follows:

“Patients with TS have increased gonadotropin concentration from infancy or older age, usually the age typical for the start of puberty in healthy girls, and low titers of estrogens. Hypergonadotropic hypogonadism is a typical TS feature in adolescence and adulthood.”

Response to Question 6

The conclusive question has been modified as follows:

“In our rare, bilateral case of PRL, beyond TS, the patient had the tendency to inflammatory-driven conditions including hypothyroidism and a suspected but yet unconfirmed autoimmune liver disorder.”

Carla Minoia on behalf of coauthors

Round 2

Reviewer 3 Report

Please see the attached review.

Author Response

30-6-21

Response to Reviewer

Dear Colleague,

Thank you for your comments and attention to the presented case report.

I tried to address your questions and modify the manuscript accordingly.

Question 1

Thank you for improving the manuscript. However, I still have remarks regarding the main issue- hypothesis linking PRL to autoimmune tendency in the presented patient with TS.

- The diagnosis “post-thyroiditis hypothyroidism” still does not give the background of autoimmune origin. Autoimmune thyroiditis is in fact not “post-thyroiditis”, as the autoimmune inflammation is usually present all along the follow-up. In your answer the information “thyroid autoimmunity: TSH 8.49 uUI, FT4 1.88 ng, FT3 1.82 ng” actually does not give the autoimmune tests results, only thyroid function results during levothyroxine treatment. The autoimmune tests include antithyroid antibodies (anti-TPO and anti-TG) and are easily available. Therefore, I would strongly recommend to present these results. Otherwise the conclusion is not supported by the results.

Response to question 1

We have performed the requested exams on the 23rd June 2021 and they resulted normal, with anti- TPO 10 U/ml and anti-Tg 12.3 U/ml. We now can conclude that the hypothyroidism post-thyroiditis is related to inflammation not of auto-immune origin.

Question 2

- Moreover, the additional information regarding hypertransaminasemia does not support the suspected diagnosis of autoimmune liver disorder. I would rather suspect nonalcoholic fatty liver disease (NAFLD) that may result e.g. from overweight/obesity. Overweight is quite common feature in women with TS, we do not have the information about the BMI value of the presented patient.

Response to question 2

Thank you for the suggestion. Unfortunately, the patient did a gastroenterological evaluation comprehensive of the autoimmune exams, but this hypothesis was not taken under consideration.

However, the Authors consider NAFLD as a potential differential diagnosis and thus it was included in the text. The patient has a BMI of 30.3 and it has also been reported in the text. Also congenital hepatic morphostructural alterations have been reported in patients with Turner syndrome. Thus, the Authors modified the text as follows:

“Patient comorbidities included hypertension, hypothyroidism post-thyroiditis, and obesity (Body Mass Index – BMI – 30.3).”

“She completed the diagnostic work-up with blood cell count, blood chemistry, and Hepatitis and HIV serology, which resulted within the normal limits, except for a known hypertransaminasemia (transaminases 2 folds higher). , but a conclusive diagnosis to explain this alteration was not possible, and a differential diagnosis between nonalcoholic fatty liver disease (NAFLD) (cholesterol 273 mg/dl, high-density lipoprotein – HDL- 54 mg/dl, obesity) and autoimmune liver disorder, though autoimmunity was negative (antinuclear antibodies - ANA, anti-smooth muscle antibodies - ASMA, anti–liver-kidney microsomal antibody – anti-LKM, anti-mithocondria antibodies – AMA) was taken into consideration. It is to be considered that Turner syndrome often manifests with hypertransaminasemia whose cause is not always documentable and could be related to morphostructural congenital hepatic alterations [14,15]. Hepatic alteration in TS most commonly manifests as an asymptomatic hypertransaminasemia and, while in many cases liver injury does not progress to cirrhosis, there is a 6-fold increased risk of cirrhosis in patients with TS compared with the general population [20,22]. “

Two more references have been added:

14 - El-Mansoury M, Berntorp K, Bryman I, Hanson C, Innala E, Karlsson A, Landin-Wilhelmsen K. Elevated liver enzymes in Turner syndrome during a 5-year follow-up study. Clin Endocrinol (Oxf). 2008;68(3):485-90. doi: 10.1111/j.1365-2265.2007.03166.x.

15 - Roulot D. Liver involvement in Turner syndrome. Liver Int. 2013 Jan;33(1):24-30. doi: 10.1111/liv.12007

Question 3

In my opinion lack of the exact data regarding hypothyroidism and hypertransaminasemia mentioned above makes it impossible to formulate the conclusion presented by Authors. On the other hand this hypothesis is possible and interesting. I would strongly recommend performing the antithyroid antibodies tests in the patient and discussing the hypothyroidism and hypertransaminasemia with more detail (as above, with the results of both ultrasound examinations). If the Authors include these data in the paper the conclusion would be reliable and important.

Response to Question 3

Due to the lack of certainty on the etiopathogenesis of the lymphoma, we modified the discussion as follows:

“In our rare, bilateral case of PRL, beyond TS, the patient had the tendency to inflammatory-driven conditions including hypothyroidism post-thyroiditis The “inflammatory nidus theory” could be thus hypothesized as the etiopathogenesis of the lymphoma in our patient, but we have not enough elements to strength this assumption. “

Thank you